# Characterization and Comparative Analysis of Chloroplast Genomes of Medicinal Herb *Scrophularia ningpoensis* and Its Common Adulterants (Scrophulariaceae)

**DOI:** 10.3390/ijms241210034

**Published:** 2023-06-12

**Authors:** Lei Guo, Xia Wang, Ruihong Wang, Pan Li

**Affiliations:** 1Zhejiang Province Key Laboratory of Plant Secondary Metabolism and Regulation, College of Life Sciences and Medicine, Zhejiang Sci-Tech University, Hangzhou 310018, China; 202230902120@mails.zstu.edu.cn (L.G.); 202220901067@mails.zstu.edu.cn (X.W.); 2Laboratory of Systematic & Evolutionary Botany and Biodiversity, College of Life Sciences, Zhejiang University, Hangzhou 310058, China

**Keywords:** *Scrophularia*, chloroplast genome, species identification, comparative genomics, phylogenomics

## Abstract

*Scrophularia ningpoensis*, a perennial medicinal plant from the Scrophulariaceae family, is the original species of *Scrophulariae* Radix (SR) in the Chinese Pharmacopoeia. This medicine is usually deliberately substituted or accidentally contaminated with other closely related species including *S. kakudensis*, *S. buergeriana*, and *S. yoshimurae*. Given the ambiguous identification of germplasm and complex evolutionary relationships within the genus, the complete chloroplast genomes of the four mentioned *Scrophularia* species were sequenced and characterized. Comparative genomic studies revealed a high degree of conservation in genomic structure, gene arrangement, and content within the species, with the entire chloroplast genome spanning 153,016–153,631 bp in full length, encoding 132 genes, including 80 protein-coding genes, 4 rRNA genes, 30 tRNA genes, and 18 duplicated genes. We identified 8 highly variable plastid regions and 39–44 SSRs as potential molecular markers for further species identification in the genus. The consistent and robust phylogenetic relationships of *S. ningpoensis* and its common adulterants were firstly established using a total of 28 plastid genomes from the Scrophulariaceae family. In the monophyletic group, *S. kakudensis* was determined to be the earliest diverging species, succeeded by *S. ningpoensis*. Meanwhile, *S. yoshimurae* and *S. buergeriana* were clustered together as sister clades. Our research manifestly illustrates the efficacy of plastid genomes in identifying *S. ningpoensis* and its counterfeits and will also contribute to a deeper understanding of the evolutionary processes within *Scrophularia.*

## 1. Introduction

*Scrophularia* Linnaeus is a genus of plants in the family Scrophulariaceae, which comprises over 250 extant species of plants that are mainly found in temperate regions of the Northern Hemisphere [1,2,3]. The dried roots of many *Scrophularia* plants have therapeutic properties and are widely used as herbal medicines around the world [4,5,6,7]. *Scrophularia ningpoensis* Hemsley, commonly known as “Xuan Shen” in Chinese, is a perennial medicinal herb endemic to China and the original species of *Scrophulariae* Radix (SR) in traditional Chinese medicine (TCM) with a domestication history of more than 1000 years [8]. The dried roots of *S. ningpoensis* are recorded as SR in the Chinese Pharmacopoeia and used to cure fever, constipation, swelling, rheumatism, and inflammatory affections [8,9]. It has been found in a previous survey that *S. ningpoensis* is commonly mixed and substituted with other adulterants in East Asia, such as *S. buergeriana* Miq., *S. kakudensis* Franch, and *S. yoshimurae* T. Yamazaki [10,11,12,13]. Studies conducted to identify the secondary metabolites in these *Scrophularia* plants have revealed marked differences in the contents and concentrations of the bioactive compounds they contain [4,14,15,16]. While they all contain iridoids, phenylethanoids, and flavonoids, there are some differences in the specific bioactive compounds [16,17,18]. The clinical safety and effectiveness of these species may be compromised due to the controversial identification caused by their similar and indistinguishable morphology, as well as the use of dried and sliced forms in medicinal material trade.

In more detail, *S. ningpoensis* is widely distributed and artificially cultivated throughout the central and eastern regions of China, and it is especially known as one of the famous authentic medicinal materials, “Zhe Ba Wei”, in Zhejiang Province [19,20,21]. *S. yoshimurae* is employed as a substitute for *S. ningpoensis* in Taiwan Province; *S. buergeriana* and *S. kakudensis* spread from northern China to Japan and Korea and have been prescribed and used as SR in Korea for a long time [13,14,15,16,17,18]. However, because of the resource scarcity of *S. buergeriana*, imported *S. ningpoensis* has also been used as a substitute in the Korean herbal market [10,11]. The development of molecular biology has made DNA barcoding a useful identification technique, which has gained popularity in herbal medicine research [22,23]. Systematic phylogenetic analysis and species identification of *Scrophularia* have also been studied using DNA barcoding, including ITS, *ndhF*, *trnQ*-*rps16*, *psbA*-*trnH*, and *trnL*-*trnF*, with some substitutes of *S. ningpoensis* not included [3,24,25,26]. Lee et al. used ITS especially to differentiate the three species (*S. buergeriana*, *S. ningpoensis*, *S. kakudensis*) utilized in *Scrophulariae* Radix [10]. Our preliminary phylogenetic analysis of the *Scrophularia* genus, which utilized universally applicable DNA barcodes (ITS, *psbA*-*trnH*, *trnL*-*trnF*), positioned *S. ningpoensis*, *S. kakudensis*, *S. buergeriana*, and *S. yoshimurae* as a monophyletic group situated within the East Asian clade but lacked phylogenetic resolution within the four closely related species [27,28]. Traditional plant classification methods based on morphological features are not always adequate for the delimitation of herbs that resemble each other. Furthermore, the use of DNA barcoding for analysis has limitations in terms of resolution, making it difficult to differentiate certain closely related or similar species. Therefore, the lack of an international industry standard or effective approach to distinguish these *Scrophularia* plants has already perplexed traders and consumers for a considerable time.

To address these methodological challenges, numerous molecular biology techniques that employ various genetic data have been developed for the purpose of species identification [29,30,31]. In comparison to conventional molecular markers, the complete plastid genome data matrix offers a greater number of genetic variation sites and can be applied in the fields of species delimitation and molecular marker development [32,33,34]. Hence, in the current study, we sequenced, assembled, and annotated the plastomes of four *S. ningpoensis* species. For the first time, the differences between *S. ningpoensis* and its common adulterants were meticulously examined and documented with the goals of (1) characterizing and contrasting the complete plastid genomes of *Scrophularia*; (2) understanding the evolutionary pattern of the plastid genome of *Scrophularia* species and resolving phylogenetic relationships within the four *Scrophularia* species; and (3) evaluating variations in high-variability sites and simple sequence repeats (SSRs) for accurate authentication of *Scrophularia* and its counterfeit herbs. Our study will contribute to the expansion of the genomic resources available for *Scrophularia* and provide critical information to support the identification and phylogenetic analysis of different *Scrophularia* species. Additionally, the research findings will also have practical applications for the safe use of *Scrophulariae* Radix (SR).

## 2. Results

### 2.1. Morphological Comparison

Our morphological study revealed the similar morphological characters and habitat preferences (forests along streams, thickets, and tall grasses) of the complex of four *S. ningpoensis* species, mainly differing in calyx lobes, inflorescences, and the serrate degree of leaf margin (Figure 1). *S. buergeriana* stands out with its yellow-to-green corolla color and subcapitate or terminal raceme/spike inflorescences, distinguishing it from the other three species. The calyx lobes of *S. ningpoensis* and *S. buergeriana* are obtuse to rounded, while *S. kakudensis* and *S. yoshimurae* have acute to acuminate lobes. Leaf characteristics also differ, with serrate margins in *S. ningpoensis*, *S. buergeriana*, and *S. kakudensis*, while *S. yoshimurae* has double-serrate margins. Leaf shapes and bases vary, ranging from ovate in *S. ningpoensis* to ovate–lanceolate in *S. yoshimurae* [35] (Figure 1, Appendix A).

### 2.2. Chloroplast Genome Structure and Characteristics Analyses

The chloroplast genomes of 5 individuals of the 4 *Scrophularia* species were sequenced using the Illumina HiSeq 2500 system, generating clean data ranging from 2.25 to 3.79 Gb. Integrating the previously published cp genome of *S. ningpoensis* (NC_053823, MN734369), *S. buergeriana* (NC_031437, KP718626), and *S. kakudensis* (MN255822), the length of the cp genomes of the *S. ningpoensis* complex were in the range of 153,016 bp (*S. kakudensis*) to 153,631 bp (*S. buergeriana*) (Table 1). The entire chloroplast genome has four standard structural regions common in angiosperms, which include a large single copy (LSC), a small single copy (SSC), and two reverse repeat regions (IRa and IRb) (Figure 2). The length of the IR region fluctuated between 25,482 bp (*S. buergeriana*, OQ633013) and 25,490 bp (*S. ningpoensis*, OQ633009); the LSC region ranged from 84,130 bp (*S. kakudensis*, OQ633011) to 84,274 bp (*S. yoshimurae*, OQ633010); and the SSC region extended from 17,922 bp (*S. kakudensis*, OQ633012 and OQ633011) to 17,938 bp (*S. ningpoensis*, OQ633009). Moreover, the GC content in the IR region was higher than in the LSC and SSC regions. The total GC content varied slightly, from 37.98% to 37.99%, the content of the LSC region varied from 36.07% to 36.08%, the content of the SSC region varied between 32.15% and 32.18%, and the content of the IR region was 43.16%~43.19%.

The 10 *Scrophularia* plastomes encoded 132 genes, namely, 80 protein-coding genes (CDS), 4 ribosomal RNA (rRNA) genes, 30 transfer RNA (tRNA) genes (totaling 114 unique genes), and 18 duplicated genes (Appendix A). Among the 114 unique genes, 10 protein-coding genes (*petB*, *petD*, *atpF*, *ndhB*(×2), *rpl16*, *rpl2*(×2), *rps16*, *rpoC1*) and 5 tRNA genes (*trnA*-*UGC*, *trnG*-*GCC*, *trnI*-*GAU*, *trnL*-*UAA*, *trnV*-*UAC*) contained a single intron, while 3 genes (*clpP*, *rps12*, and *ycf3*) had 2 introns. The *ycf1* gene in this region has been identified as a pseudogene in nearly all currently sequenced plant chloroplast genomes [36], and its coding sequence contains internal stop codons, resulting in high variability. Additionally, the partial duplication of *rps19* at the LSC/IRa junction region has lost its protein-coding capability as a result of the incomplete duplication. Our results confirmed that the ten *Scrophularia* chloroplast genomes have maintained a high degree of conservation in genomic characteristics, such as gene content and arrangement and GC content throughout the process of evolution.

### 2.3. Contraction and Expansion of Inverted Repeats

The comparison of overall sequence variation indicated that the IR region was highly conserved, but there were differences in the IR/SC boundary regions across various plant species. In the case of the *rps19* gene, it spanned the LSC/IRb boundary and primarily occurred in the LSC region, covering a length of 238 bp, with the remaining 41 bp located in the IRb region, whereas in *S. kakudensis* (MN255822) it comprised 239 bp situated in the LSC and 40 bp in the IRb region. At the IRa/LSC boundary, the *rps19* gene extended 2 bp to the LSC region which caused the *rps19* pseudogene fragment to be positioned at the boundary, while in *S. kakudensis* (MN255822) it extended only 1 bp. However, in *S. ningpoensis* (OQ633009) the *rps19* genes did not expand to the LSC region. The *ycf1* gene located in the SSC/IRa boundary had 4485 bp in the SSC and 882 bp in the IRa region, whereas in *S. kakudensis* (OQ633012) it had 4467 bp situated in the LSC and 882 bp in the IRa region. Of these species, the distance from *ycf1* to the IRb/SSC boundary was 1 bp in *Scrophularia* with GenBank accession numbers of MN734369, NC_053823, NC_031437, KP718626, and MN255822 and 5 bp in OQ633009~OQ633013 (Figure 3). In general, the differences in length of the cp genomes among the four *Scrophularia* species can be attributed to the expansion and contraction of the IR/SC boundary regions. By using Mauve software, collinearity analysis was conducted on the chloroplast genomes of the four *Scrophularia* species, which revealed that the genomes exhibited strong collinearity, characterized by a single-color block, conserved genes, and homologous regions arranged in a linear manner, with no instances of gene rearrangement or inversion (Appendix A).

### 2.4. Codon Usage Analysis

The analysis of codon bias revealed that different *Scrophularia* species exhibit variation in the number of codons and the effective number of codons. The average number of codons for *Scrophularia* individuals was 51,058 (*S. ningpoensis*), 51,156 (*S. buergeriana*), 51,007 (*S. kakudensis*), and 51,054 (*S. yoshimurae*), while the average effective number of codons (ENC) was 55.83 (*S. ningpoensis*), 55.91 (*S. buergeriana*), 55.71 (*S. kakudensis*), and 56.13 (*S. yoshimurae*) (Appendix A). The relative synonymous codon usage (RSCU) value is the relationship between the actual and expected occurrences of a codon [37]. Among the 64 codons that encode amino acids, except for the 3 stop codons and the unbiased methionine (Met) and threonine (Thr) (RSCU = 1), 27 codons exhibited a preference with RSCU values greater than 1, indicating that these codons had higher priority. Among these, the AGA codon corresponding to arginine (Arg) had the highest recognition rate, with an average RSCU of 1.92. The remaining 32 codons showed low bias, with RSCU values less than 1 (Appendix A). In addition, there was a strong correlation between the codon bias and the GC content of the third position of synonymous codons. Among the codon usage of all species, the total GC content was about 37.99% or 37.98%, and the average GC content of the third base of synonymous codons ranged from 37.58% (*S. kakudensis*) to 38.51% (*S. buergeriana*), indicating that the A/U-ending codons were more preferred. These results suggested that there was a slight bias in the codon usage among the four *Scrophularia* species cp genomes.

### 2.5. Repeat Sequence Analysis

Repeat sequence analysis in the chloroplast genome is a useful tool for understanding the structure, function, and evolution of this important organelle in plants. The analysis using REPuter demonstrated that the 10 genomes presented 4 types of dispersed repeat sequences, including forward (F), reverse (R), palindromic (P), and complementary (C) repeats. A total of 454 repeats were detected, consisting of 216 F, 227 P, 9 R, and 2 C repeats. F and R repeats were the most common types (Figure 4A). *S. buergeriana* (OQ633013, NC_031437, KP718626) possessed the highest count of repeats (47), while *S. kakudensis* (OQ633011, MN255822) had the lowest (44). We calculated the frequencies of different repeat sequence lengths in 10 individuals of *Scrophularia*, and the proportion of 30–39 bp repeat sequences reached 74.45% (Figure 4B). According to the principle that the identical-length fragments found in homologous DNA regions are deemed to share repetitive sequences [38,39], 35 repetitive sequences were identified in the 10 *Scrophularia* chloroplast genomes. In addition, *S. buergeriana* (OQ633013, NC_031437, KP718626) possessed the highest number of repetitive sequences (12), whereas *S. kakudensis* (OQ633011, MN255822) exhibited the lowest number of repetitive sequences (9) (Appendix A).

### 2.6. Simple Sequence Repeat Analysis

Simple sequence repeats (SSRs), also known as microsatellites, are molecular markers that consist of short, repetitive DNA sequences that are tandemly repeated [40,41]. The chloroplast genomes of 10 *Scrophularia* species were analyzed by Misa, and a total of 422 SSRs were detected across the 10 individuals. The highest number of SSRs was observed in *S. ningpoensis* (OQ633009) (44), while the lowest number was found in *S. yoshimurae* (39) (Appendix A). Both *S. buergeriana* and *S. kakudensis* (OQ633011, MN255822) had 42 SSRs, but the number of SSRs in *S. kakudensis* (OQ633012) was slightly lower at 41 (Appendix A). Most of these SSRs (94.55%) were composed of A/T base components. Out of the SSRs analyzed, a total of 339 single nucleotides were identified in 10 *Scrophularia* species, which accounted for 80.33% of the SSRs. All 37 dinucleotides detected were of the AT/TA type, making up 8.77% of the SSRs. Additionally, there were 10 trinucleotides of the AAT/ATT type, accounting for 2.37% of the SSRs, and 33 tetranucleotides with diverse patterns (AAAG/CTTT, AAAT/ATTT, AATC/ATTG, ACAG/CTGT), which represented 7.82% of the SSRs. Finally, there were only 3 pentanucleotides, all of which were AAAAT/ATTTT-type, making up 0.71% of the SSRs. The repeated sequences in the form of AT/TA were present in all four *Scrophularia* species, but only *S. kakudensis* had repeats in the form of TTTTA, whereas the other three species did not (Figure 5A). Through analysis, we also found that most SSR loci were located in IGS (45.97%), followed by introns (35%) and CDS (18.96%) (Figure 5B), which may be due to the high mutation rate of the intergenic spacers (IGS).

### 2.7. Selective Pressure Analyses

We used *Scrophularia takesimensis* (GenBank acc. no. KP718628) as the reference sequence. The nonsynonymous (Ka) and synonymous (Ks) substitutions of 10 chloroplast genomes were calculated using TBtools V1.113. The Ka/Ks (ω) values were used to analyze the selection pressure of 80 protein-encoding genes in *Scrophularia*. The highest Ka/Ks value was 1.4207 in *S. buergeriana* (OQ633013, NC_031437, KP718626) (Appendix A). The Ka/Ks of most genes in *Scrophularia* individuals were less than 1, indicating that the genome species had high evolutionary constraints, and most of the genes had undergone purification selection [34]. A few genes of *Scrophularia* had Ka/Ks greater than 1, such as the *rps12* gene of *S. ningpoensis* (MN734369, NC_053823) and the *ycf2* gene of 7 individuals of 3 species of *S. ningpoensis* (OQ633009, MN734369, NC_053823), *S. buergeriana* (OQ633013, KP718626, NC_031437), and *S. yoshimurae* (OQ633010) (Figure 6). The *ndhF* genes of five individuals from three species of *S. ningpoensis*, *S. buergeriana* (OQ633013), and *S. yoshimurae* (OQ633010) showed that these sites evolved rapidly and there may be positive selection sites.

### 2.8. Sequence Divergence Analysis

mVISTA was used to compare the differences among 10 cp genomes, with *S. buergeriana* (NC_031437) serving as the reference genome and DnaSP used to analyze the nucleic acid diversity and identify divergence hotspot regions of the 10 *Scrophularia* chloroplast genomes. According to the results obtained from the mVISTA analysis, these species exhibited a high degree of sequence similarity (Figure 7). The results showed that the *Pi* values of nucleotide diversity of these 10 species ranged from 0 to 0.01115 (Appendix A). We detected 8 intergenic spacers (IGSs), with high divergence values (*Pi* > 0.004), namely, *trnH*-*GUG*-*psbA*, *psbZ*-*trnG*-*GCC*, *trnT*-*UGU*-*trnL*-*UAA*, *trnP*-*GAA*-*ndhJ*, *ndhJ*-*ndhk*, *rbcL*-*accD*, *psbE*-*petL*, and *psaC*-*ndhE*. The sequence of the *ndhF* gene had high nucleotide diversity with a *Pi* value of 0.422. In the chloroplast genome, the sequence divergence observed in the IR regions was comparatively lower than that of the LSC and SSC regions, and the coding region was found to be more conserved than the non-coding regions. The outcomes obtained from the mVISTA and DnaSP analysis were consistent with each other. By analyzing the divergence hotspots, we were able to identify the dissimilarities between related species (Figure 8). The divergence hotspot regions identified in the LSC and SSC regions can be utilized to discern and estimate the divergence among closely related species.

### 2.9. Phylogenetic Analysis

The phylogenetic reconstruction based on 28 complete cp genomes of Scrophulariaceae was performed using the maximum likelihood method (ML) and Bayesian inference analysis (BI). The phylogenetic trees of the two methods produced almost the same tree topology and were integrated together with high support values of the BI posterior probability (PP) and bootstrap support values of the ML labeling in each branch. Almost all of the nodes in the phylogenetic tree received strong support (PP/BS = 1/100). The phylogenetic tree had three monophyletic groups, including tribe *Scrophularieae*, tribe *Budlejeae*, and tribe (*Myoporeae* + *Leucophylleae*) as outgroups. *Buddleja* was the sister of *Scrophularia* and *Verbascum*. The genus *Scrophularia* was divided into two parts: *S*. sect. *Caninae* and *S.* sect. *Scrophularia*. In this part of *S*. sect. *Scrophularia*, we analyzed 10 chloroplast genomes of the monophyletic *S. ningpoensis* complex. *S. kakudensis* was the earliest branch to be differentiated, followed by *S. ningpoensis*, *S. buergeriana*, and *S. yoshimurae* clustered into a sister branch (Figure 9).

## 3. Discussion

The chloroplast genome serves as a valuable resource for investigating plant evolution, phylogenetics, and genetic diversity [42,43,44]. *Scrophularia* comprises more than 250 species; despite the diverse range of species in this genus, only a limited number of chloroplast genome sequences have been deposited into GenBank [32,45]. In particular, the *S. ningpoensis* complex is a group of perennial herbs with pharmaceutical properties that have been applied in traditional Asian medicine for hundreds of years [4,8]. This study reports the complete chloroplast genomes of four *Scrophularia* species, including *S. ningpoensis*, *S. buergeriana*, *S. kakudensis*, and *S. yoshimurae*, among which the *S. yoshimurae* cp genome is reported for the first time, while the cp genomes of *S. ningpoensis*, *S. buergeriana*, and *S. kakudensis* have been reported before, but this is the first systematic comparison and analysis of the cp genomes of all four species. The comparative analysis revealed that the chloroplast genome structure and features were highly conserved relative to nuclear and mitochondrial genomes within the four species. The determined nucleotide sequences of the 4 species’ chloroplast genomes ranged narrowly from 153,016 bp to 153,631 bp, with a similar gene pattern of 132 genes, which has been observed in other *Scrophularia* plants (Figure 2, Table 1) [32,45]. The LSC region contains most of the genes required for chloroplast function, including those involved in photosynthesis, carbon fixation, and amino acid synthesis; the SSC region contains a smaller number of genes, including those involved in RNA processing and protein synthesis; the IR regions contain genes involved in photosynthesis, as well as rRNA genes and tRNA genes [46,47,48]. The observed conservation of the *S. ningpoensis* complex’s chloroplast genomes can be attributed to the similarities in the morphological characters and habitat preferences, as well as the low rate of chloroplast substitution [27,45].

Inverted repeats (IRs) are regions of the chloroplast genome where the DNA sequence is duplicated and inverted, resulting in two copies of the same sequence in opposite orientations [43]. The contraction and expansion of IR regions can have significant impacts on the evolution of the chloroplast genome and the genes that are located within them [49,50]. While expansion can lead to functional divergence or even the evolution of new genes, contraction can cause the pseudogenization of genes and a lack of genetic information [50,51,52]. Pseudogenization has been observed in various organisms and can have significant implications for their evolutionary trajectories and adaptation to changing environments [53,54,55]. The *ycf* gene is an essential open reading frame present in the cp genome of all photosynthetic organisms and highly conserved across different species; it plays a fundamental role in the functioning of the chloroplast, though the exact function and regulation of the gene are still not fully understood [56,57,58]. Comparative genomic studies have revealed that the *ycf1* pseudogene is present in nearly all plant cp genomes sequenced to date and exhibits significant variability [59,60]. The IR/LSC junctions of *Scrophularia* extended into the *rps19* gene, which was consistent with the chloroplast genome characteristics of many typical angiosperms [36]. The *rps* gene is an important gene in the chloroplast genome, coding for the ribosomal protein S which is critical for protein synthesis. Its conservation across photosynthetic organisms indicates its importance, while its evolution highlights the dynamic nature of the chloroplast genome [61]. Further research on the *rps* gene and its function could provide valuable insights into the evolution of photosynthetic organisms and their ability to adapt to changing environments.

SSR markers have been widely utilized in genetic research due to their high level of polymorphism, co-dominant inheritance, and ease of detection [62,63]. They are useful for genetic diversity studies, population genetics, and linkage mapping and are relatively inexpensive and accessible to researchers with limited resources [64]. We detected 39 to 44 cp SSR markers (Appendix A) in the 10 individuals, which can be employed for further population genetic research in *Scrophularia*. Codon usage bias is the phenomenon of the unequal usage of synonymous codons in the genetic code of an organism [65,66]. In other words, certain synonymous codons are utilized more frequently than others to code for the same amino acid in the protein-coding genes. Leucine (Leu) and Tryptophane (Trp) were the most prevalent amino acids in *Scrophularia*, with an average count of 5137 and 705, respectively, which accords with previous studies on *Scrophularia* species. Meanwhile, the codons that terminate with A and/or T (U) typically exhibited high RSCU scores across the 10 cp genomes, e.g., AGA (1.97) for Arginine, GAU (1.47) for Aspartate, and CUU (1.31) for Leucine, which is largely due to a pronounced preference for the synonymous codons ending in A and T codons. Understanding codon usage bias can be important for optimizing gene expression in genetic engineering and synthetic biology, as well as for understanding the evolutionary history and biology of organisms [67].

Understanding the selective pressures acting on genes can provide insights into their function, adaptation, and diversity [68]. One of the ways to measure the selective pressures acting on genes is to compare the frequencies of synonymous and nonsynonymous substitutions [69]. In this study, the selective pressure analysis performed on the genes revealed that the majority of the genes exhibited a Ka/Ks ratio of less than 1 and had undergone purifying selection, indicating their functional and evolutionary importance as the target of natural selection. The finding that most genes had undergone purifying selection is not surprising as it is essential for maintaining the function and stability of genes. Nucleotide variability of different individuals or populations is typically measured using a statistic called the *Pi* value, which is calculated by comparing DNA sequences in the sample and counting the number of nucleotide differences between them, and then dividing this number by the total number of sites compared [39,70]. In the present investigation, we detected 8 intergenic spacers (IGSs) with elevated divergence values (*Pi* > 0.004) that could be used to infer evolutionary relationships among different groups of *Scrophularia*.

Phylogenetic analysis is the study of the evolutionary relationships among organisms based on their genetic, morphological, or other characteristics. It can be performed using diverse methods, such as ML, BI, and maximum parsimony (MP) [71,72,73]. Phylogenetic analysis has broad applications in fields such as evolutionary biology, systematics, biogeography, conservation biology, and comparative genomics, and it has enabled scientists to gain a better understanding of the history of life on earth and the diversity of organisms that inhabit our planet [74]. Our study presents the first robust and consistent phylogenetic relationships of the *S. ningpoensis* complex, utilizing a substantial sampling of 28 complete chloroplast genomes from 19 species of Scrophulariaceae based on ML and BI. Three individuals of *S. ningpoensis*, *S. buergeriana*, and *S. kakudensis* as biological duplicates were taken to improve the reliability of the data, excepting *S. yoshimurae* with a narrow distribution in Taiwan Province. Both the ML and BI analyses indicated that *Scrophularia* can be separated into two monophyletic lineages: *S.* sect. *Caninae* and *S.* sect. *Scrophularia* (Figure 9). The *S. ningpoensis* complex clustered into a strongly supported monophyletic clade within *S*. sect. *Scrophularia*. The species of *S. kakudensis* from Korea and Liaoning Province in northern China clustered together and was determined as the earliest diverging lineage at the basal position, followed by the *S. ningpoensis* lineage from eastern China. The *S. buergeriana* lineage from Korea and Hebei Province in northern China established sister relationships with *S. yoshimurae* from Taiwan Province. This research not only provides a clearer picture of *Scrophularia*’s evolutionary history but also offers important genetic information on the chloroplasts that will be useful in investigating the origins and biodiversity patterns within the Scrophulariaceae plant family. In conclusion, by conducting a comprehensive phylogenetic analysis using chloroplast genomic data, we have made significant strides in understanding the intricate relationships within the *Scrophularia* genus, ultimately contributing valuable insights into the evolutionary history of Scrophulariaceae as a whole.

## 4. Materials and Methods

### 4.1. Phenotype Observations and Plant Samples

The four *Scrophularia* species were subjected to observation of their morphological features in accordance with the general plant classification methods and the taxonomic characteristics of *Scrophularia* spp. in Flora of China [35]. All the morphologic records were based on our field observations. Leaves of the five individuals of main clades of the *S. ningpoensis* complex, including two individuals of *S. kakudensis* from Liao Ning Province, *S. ningpoensis* from Zhejiang Province, *S. buergeriana* from Hebei Province, and *S. yoshimurae* from Taiwan Province, were collected. Voucher specimens were deposited in the herbarium of Zhejiang University (HTU, Hangzhou, China). Complete cp genomes sequences of *S. ningpoensis* (NC_053823, MN734369), *S. buergeriana* (NC_031437, KP718626), and *S. kakudensis* (MN255822) were obtained from NCBI for the purpose of comparison with the *S. ningpoensis* complex.

### 4.2. Genomic DNA Extraction, Sequencing, Assembly, Annotation, and Submission to GenBank

A DNA library, including original reads, was constructed by randomly interrupting total DNA with ultrasound. These original reads were assembled into complete plastid genomes by de novo assembly and reference-guided combination [75]. First, the modified CTAB method was used to extract DNA from plant leaves. Genomic DNA was extracted from silica-dried leaf samples using the DNA Plantzol kit (Invitrogen, Carlsbad, CA, USA). Chloroplast sequences were then produced using the platform Illumina HiSeq2500 at the Beijing Genomics Institute (BGI, Shenzhen, China). All raw readings were processed using Trimmomatic v0.39 software (Julich, Germany) to remove adapter sequences, short reads (length: 75 bp), and low-quality bases (q value: 20). Subsequently, homologous plasmid sequences were chosen from complete cellular DNA reads, and a de novo assembly pipeline was used to assemble these sequences into numerous contigs. Because of the organelles and nuclear DNA included in the sequencing data [32], GetOrganelle was utilized to assemble the complete chloroplast of *Scrophularia* to filter contigs. Finally, the filtered contigs were ordered and aligned to the chloroplast genomes of related species that have been previously published, using the reference chloroplast genome (MN734369) as a guide. The chloroplast genome was annotated by Geneious software (Geneious Biologics 2023 (https://www.geneious.com/biopharma, accessed on 10 September 2022)), and the annotation of *rps12* gene was examined with CPGview [76]. Ultimately, the five new sequencing columns were submitted to the GenBank database.

### 4.3. Contraction and Expansion of Inverted Repeats

The length variation of plant chloroplast genomes is mainly caused by the expansion and contraction of the IR region and single copy boundary [77,78]. The IRSCOPE online website (Helsinki, Finland) (https://irscope.shinyapps.io/irapp/, accessed on 7 January 2023) was used to compare IR boundary amplification and contraction of the ten sequences [79]. The Java environment was configured, and the chloroplast genome sequence in fasta format was uploaded to the Mauve [80] to analyze the collinearity of 10 *Scrophularia* chloroplast genomes.

### 4.4. Codon Usage Bias Analysis

Different organisms have different frequencies of using degenerate codons in the process of translation, and a set of common codons corresponding to them have come into being in the course of evolution [81]. The CUSP and CHIPS plugins in EMBOSS [82] (EMBOSS Explorer (inra.fr)) and CodonW V1.4.2 (https://codonw.sourceforge.net/, accessed on 9 February 2023) were used to calculate codon usage bias parameters, including the values of the effective number of codons (ENC) and relative synonymous codon usage (RSCU) and the GC content relevant to the first, second, and third codon positions in the target coding sequence (CDS) and the entire plastid [83,84].

### 4.5. Repeat Sequence Analysis

The importance of repeat sequences in phylogenetic studies and genome rearrangement is well known [85]. We used the online tool REPuter (https://bibiserv.cebitec.uni-bielefeld.de/reputer, accessed on 14 February 2023) to identify scattered repeats in the chloroplast genome, including forward repeats, reverse repeats, palindromic repeats, and supplementary repeats [86]. The parameters were set to a Minimal Repeat size of 30 bp, a Hamming Distance of 3, and a Maximum Computed Repeats of 80 times.

### 4.6. Simple Sequence Repeat Analyses

MISA [87] (http://pgrc.ipk-gatersleben.de/misa, accessed on 18 February 2023) was used to analyze simple sequence repeats (SSRs) in the chloroplast genome of *Scrophularia*. The parameters were set to single nucleotide, dinucleotide, trinucleotide, tetranucleotide, pentanucleotide, and hexanucleotide SSRs with more than 10, 6, 4, 3, 3, and 3 repeat units.

### 4.7. Selective Pressure Analyses

In genetics, the ratio of nonsynonymous (Ka) to synonymous (Ks) substitution rates (ω) is frequently used to evaluate the presence of selection pressure on protein-coding genes [88]. In this study, we used a Perl script to extract protein-coding genes from the chloroplast genome of *Scrophularia*, used Geneious to extract CDS, and calculated Ka, Ks, and Ka/Ks values for each plastid gene with TBtools V1.113 [89]. The evolutionary process of genes, i.e., positive selection, purified selection, or neutral selection, is Ka > Ks (ω > 1), Ka < Ks (ω < 1), or Ka = Ks (ω = 1), respectively.

### 4.8. Sequence Divergence Analysis and Nucleotide Diversity Analysis

To compare the chloroplast genomes among the four *Scrophularia* species, the mVISTA program (Berkeley, CA, USA) (https://genome.lbl.gov/vista/mvista/submit.shtml, accessed on 20 July 2022) was used for genome comparison. The LAGAN mode was employed with default parameters to align the ten chloroplast genomes [90]. The gene, exon, UTR, CNS, and mRNA of these sequences were compared using the mVISTA for complete sequence alignment with settings of 70% and window length of 100 bp.

In order to distinguish the hypervariable regions of the chloroplast genome of *Scrophularia*, all plastid sequences were aligned and DnaSP v6.0 [91] was used for diversity analysis. *S. buergeriana* (NC_031437) was taken as a reference, and DnaSP was used to identify the divergence hotspots of 10 *Scrophularia* chloroplast genomes. The window length was set to 600 bp and the step size was 200 bp. The total number of mutations (Eta) and nucleotide diversity (*Pi*) values of the chloroplast genome of *Scrophularia* were evaluated.

### 4.9. Phylogenetic Analysis

A total of 28 chloroplast genomes of 7 genera in Scrophulariaceae were selected for phylogenetic tree construction (Appendix A). Among them, there were 19 species in *Scrophularia*, including 2 *Verbascum*; 2 *Buddleja*; 2 *Myoporum*; 1 each for *Diocirea*, *Eremophila*, and *Leucophyllum*; and a total of 9 individuals as outgroup. We used MAFFT V7 [92] to compare these 28 plastid sequences under default parameters. The maximum likelihood method was used to construct the phylogenetic tree using IQ-TREE V1.6.8 [93], and the model was TVM+F+R2. In jModelTest v2.1.10 [94], the TVM+I+G substitution model was determined according to the Bayesian Information Criterion (BIC), and the phylogenetic tree was constructed by the Bayesian method in MrBayes V3.2.7 [95]. Monte Carlo Simulation and Markov Chain (MCMC) algorithm were used for 2 million generations. The tree was sampled every 500 generations, and the checkfreq was 5000 times. The first quarter of the calculated tree was discarded, and the Bayesian phylogenetic tree was constructed from the remaining trees [32].

## 5. Conclusions

In this study, 10 chloroplast genomes of *Scrophularia* from 4 species were analyzed. The results showed that the chloroplast genomes of *Scrophularia* were relatively conservative, but there were eight (*trnH*-*GUG*-*psbA*, *psbZ*-*trnG*-*GCC*, *trnT*-*UGU*-*trnL*-*UAA*, *trnP*-*GAA*-*ndhJ*, *ndhJ*-*ndhk*, *rbcL*-*accD*, *psbE*-*petL*, and *psaC*-*ndhE*) intergenic spacers and one (*ndhF*) protein-coding gene that had higher variation. Morphological analysis and evolutionary analysis of the 10 *Scrophularia* individuals distinguished *S. ningpoensis* from *S. kakudensis*, *S. buergeriana*, and *S. yoshimurae*, which provides support for the species identification and differentiation of Chinese herbal medicine *Scrophulariae* Radix. This study expands the genomic resources of *Scrophularia* species, presents key information for ensuring the safety and efficacy of clinical drugs, and contributes to the bioprospecting and conservation research of *Scrophularia*.

## Figures and Tables

**Figure 1 ijms-24-10034-f001:**
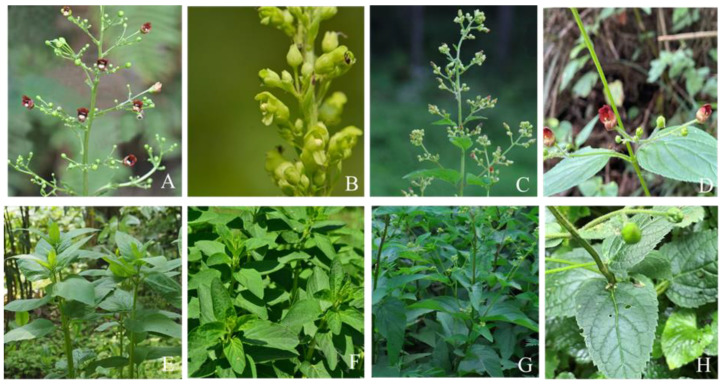
Morphological characters of four *Scrophularia* species. (**A**,**E**) Flowers and leaves of *S. ningpoensis*, (**B**,**F**) flowers and leaves of *S. buergeriana*, (**C**,**G**) flowers and leaves of *S. kakudensis*, and (**D**,**H**) flowers and leaves of *S. yoshimurae*, respectively.

**Figure 2 ijms-24-10034-f002:**
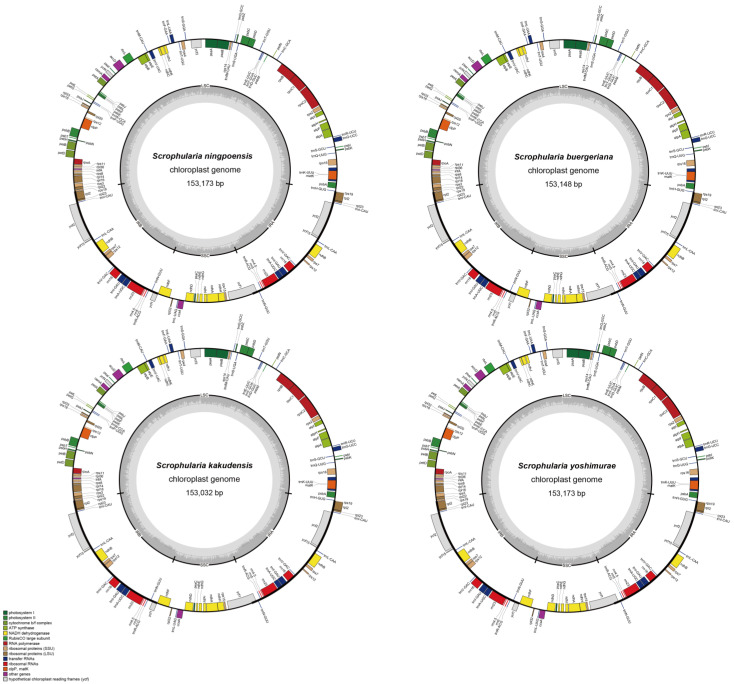
The genetic physical maps of four species, *S. kakudensis*, *S.ningpoensis*, *S. buergeriana*, and *S. yoshimurae*. Genes on the outside of the circle are transcribed clockwise, while those in the inner circle are transcribed counterclockwise. The dark gray inside represents GC content and the light gray represents AT content. The function of genes is color-coded.

**Figure 3 ijms-24-10034-f003:**
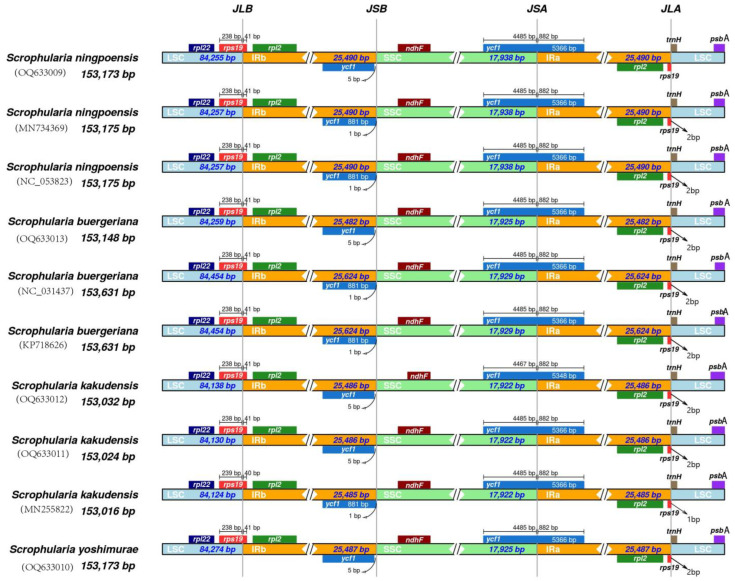
Comparison of LSC, IR, and SSC junction position among ten individuals from four *Scrophularia* species in cp genomes. JLB means the LSC/IRb junction, JSB means the SSC/IRb junction, JLA means the LSC/IRa junction, and JSA means the SSC/IRa junction.

**Figure 4 ijms-24-10034-f004:**
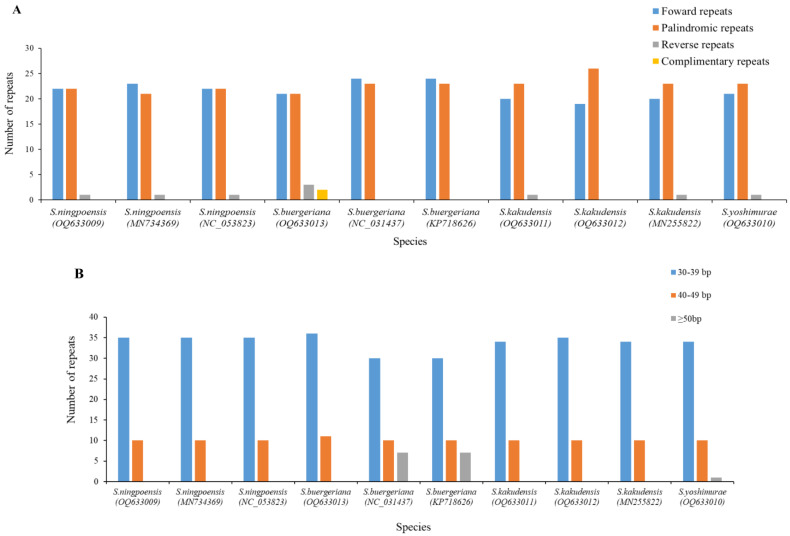
Repeat analyses in ten chloroplast genomes of *Scrophularia*. (**A**) Frequency of repeat types. (**B**) Frequency of repeats by length.

**Figure 5 ijms-24-10034-f005:**
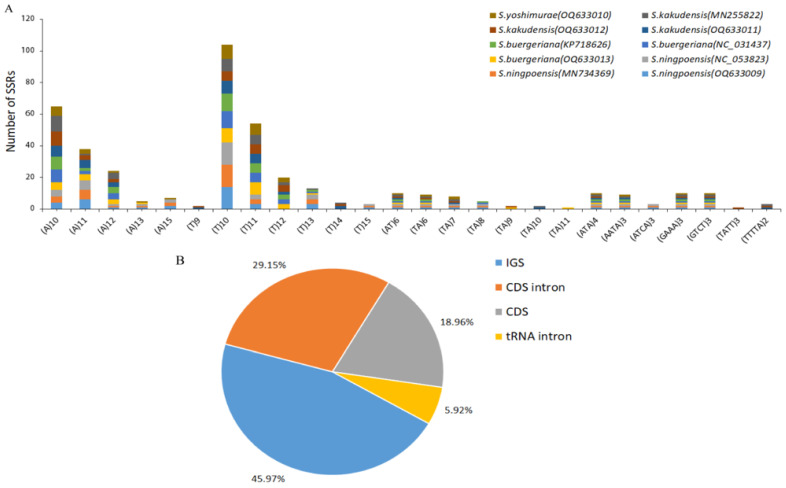
Simple sequence repeats (SSRs) in the ten *Scrophularia* chloroplast genomes. (**A**) Number of SSRs by length. (**B**) Distribution of SSR loci. CDS, coding DNA sequence; IGS, intergenic spacer region.

**Figure 6 ijms-24-10034-f006:**
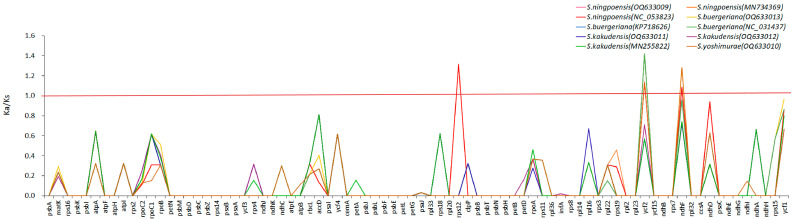
The 80 CDS regions of 10 individuals of *Scrophularia* Ka/Ks analysis.

**Figure 7 ijms-24-10034-f007:**
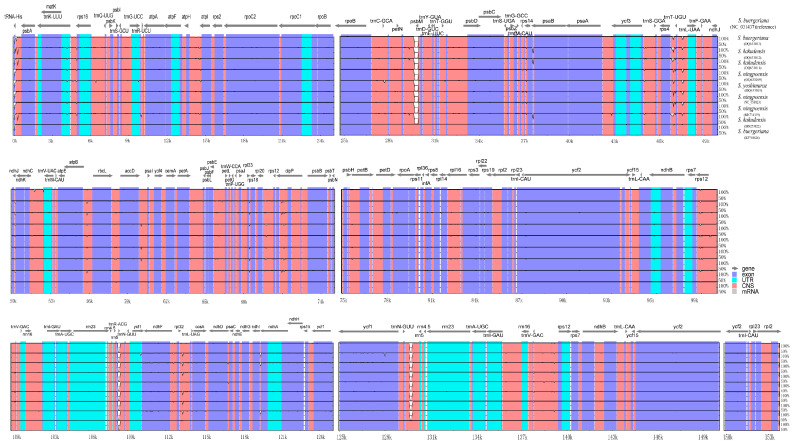
Sequence identity plots among the 10 *Scrophularia* chloroplast genomes, with *S. buergeriana* (NC_031437) as a reference. Annotated genes are displayed along the top. The vertical scale represents the percent identity between 50 and 100%. Window length of 100 bp and 70% as criteria. Genome regions are color-coded as exon, intron, and conserved non-coding sequences (CNS).

**Figure 8 ijms-24-10034-f008:**
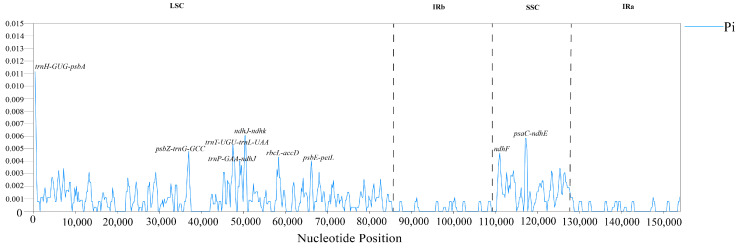
The nucleotide variation (*Pi*) values of 10 *Scrophularia* genomes were compared, with nucleotide diversity at the midpoint of the window on the x axis and within each window on the y axis (window length: 600 bp, step length: 200 bp).

**Figure 9 ijms-24-10034-f009:**
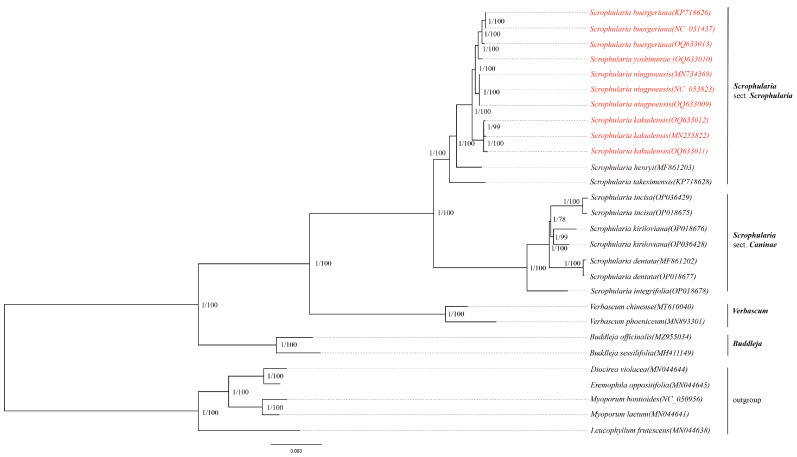
Phylogenetic relationships of ten *Scrophularia* species inferred from Bayesian inference (BI) and maximum likelihood (ML) based on the complete cp genome sequence dataset. Support values marked above the branches follow the order PP (posterior probability)/BS (bootstrap support).

**Table 1 ijms-24-10034-t001:** The basic characteristics of *Scrophularia* chloroplast genome.

Characteristics	*S. ningpoensis*			*S. buergeriana*			*S. kakudensis*			*S. yoshimurae*
**GenBank** **Acc. No.**	OQ633009	NC_053823	MN734369	OQ633013	NC_031437	KP718626	OQ633012	OQ633011	MN255822	OQ633010
**Total cpDNA size (bp)**	153,173	153,175	153,175	153,148	153,631	153,631	153,032	153,024	153,016	153,173
**LSC length**	84,255	84,257	84,257	84,259	84,454	84,454	84,138	84,130	84,124	84,274
**SSC length**	17,938	17,938	17,938	17,925	17,929	17,929	17,922	17,922	17,922	17,925
**IR length**	25,490	25,490	25,490	25,482	25,624	25,624	25,486	25,486	25,485	25,487
**Total GC content (%)**	37.99	37.99	37.99	37.98	37.99	37.99	37.98	37.98	37.98	37.98
**LSC**	36.08	36.08	36.08	36.08	36.07	36.07	36.07	36.08	36.08	36.07
**SSC**	32.18	32.18	32.18	32.17	32.17	32.17	32.15	32.16	32.17	32.18
**IR**	43.19	43.19	43.19	43.18	43.18	43.18	43.16	43.17	43.17	43.17
**Total number of genes**	132	132	132	132	132	132	132	132	132	132
**Protein-coding genes**	80	80	80	80	80	80	80	80	80	80
**rRNA genes**	4	4	4	4	4	4	4	4	4	4
**tRNA genes**	30	30	30	30	30	30	30	30	30	30
**Duplicated genes**	18	18	18	18	18	18	18	18	18	18

## Data Availability

The chloroplast genome sequences of this study are openly available at the NCBI database (https://www.ncbi.nlm.nih.gov).

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
