# Peer review of "Characterization and Comparative Analysis of Chloroplast Genomes of Medicinal Herb Scrophularia ningpoensis and Its Common Adulterants (Scrophulariaceae)"

_ijms, 2023, doi:10.3390/ijms241210034_

Round 1
Reviewer 1 Report
Authors have constructed DNA libraries of the chloroplast genomes of four species of Scrophularia species by randomly interrupting total DNA with ultrasound and assemblage of fragments into complete plastid genomes by available software programs.
The manuscript is purely descriptive without significant novelties and long paragraphs irrelevant to the molecular approach in the title, and descriptions repeated in results and discussion. Accordingly, Introduction and Discussion sections must be drastically shortened. In addition, some words are not used adequately.
1. Introductory lines 33 to 65 on geobotanic and trade of Scrophularia species should be reduced to no more than five informative lines to introduce molecular biologists into the field.
2. Similarly, the Section 2.1, on Morphological Comparison, is plenty of details that should be suppressed with appropriate references.
3. Table 1 and Figures 1-5 should be at supplementary material. In addition, lettering of Figures must be amplified (I need to guess them in my copy).
4. Line 275 (Section 2.7). 10 genomes are low number to rend significant Ka/Ks values of genes. Compare with other plant chloroplast sequences.
5. Low quality of Figure 9. Lettering small too.
6. Avoid common place rhetoric like “Chloroplasts is one of the most important organelles in plant cells” (line 334) and “Genes are the fundamental units of heredity, and the evolution of genes is crucial for the evolution of organisms” (lines 402-403).
7. Line 448 (Section 4.1). There is no “phenotype measurement” in the text. Suppress measurement.
I have no other objection on the methods and the interpretation of results. However, note that the validity of the results is based on the proper use of the software programs and the absence of significant differences in respect to previously sequenced chloroplast DNAs of closely related plants. The consequent problem is the absence of novelty suitable for the IJMS.
Reviewer 2 Report
The authors submit a morphological and a molecular comparative analysis of the chloroplast genome belonging to four Scrophularia species, of which S. ningpoensis is of particular interest due to its employment for the preparation of Scrophularia Radix, a traditional Chinese herbal remedy. The comparative analysis allowed to validate the distinctiveness of the four species based on morphological traits and clarify their reciprocal phylogenetic relationships.
The manuscript is well-written and the analyses performed are well-described. The experimental design is set properly. As stated by the authors, the processing of molecular information from a complete plastid genome may offer a greater insight on genetic variation study and can be applied in the fields of species delimitation in comparison to conventional molecular markers. The discussion is adequate to the prefixed target and the introductory remarks to the specific topics therein may help the non-specialist reader. The literature is supportive of the entire structure of the work. The reviewer has no concern about overall research quality and recommends publishing the article after taking into consideration the following few editing work:
-line 121: add “to” before “153,631bp”,
-line 166: add “was” after “boundary”,
-line 312: “species” should be substituted by “genomes”,
-line 316: remove “of” after “based on”,
-line 324: correct “Verbasum” in “Verbascum”,
-line 335: add “as” after “serves”,
-line 558: add “that” after “gene”.

Round 2
Reviewer 1 Report
The last version of the manuscript satisfactorily addresses most of my concerns to the former version. I have no additional criticism.